# Alternative Healthy Eating Index-2010 and Incident Non-Communicable Diseases: Findings from a 15-Year Follow Up of Women from the 1973–78 Cohort of the Australian Longitudinal Study on Women’s Health

**DOI:** 10.3390/nu14204403

**Published:** 2022-10-20

**Authors:** Hlaing Hlaing-Hlaing, Xenia Dolja-Gore, Meredith Tavener, Erica L. James, Alexis J. Hure

**Affiliations:** 1School of Medicine and Public Health, University of Newcastle, Callaghan, Newcastle, NSW 2308, Australia; 2Hunter Medical Research Institute, New Lambton Heights, Newcastle, NSW 2305, Australia

**Keywords:** Alternative Healthy Eating Index-2010, non-communicable diseases, multimorbidity, childbearing age, women

## Abstract

Non-communicable diseases (NCDs) and multimorbidity (≥two chronic conditions), are increasing globally. Diet is a risk factor for some NCDs. We aimed to investigate the association between diet quality (DQ) and incident NCDs. Participants were from the Australian Longitudinal Study on Women’s Health 1973–78 cohort with no NCD and completed dietary data at survey 3 (2003, aged 25–30 years) who responded to at least one survey between survey 4 (2006) and survey 8 (2018). DQ was measured by the Alternative Healthy Eating Index-2010 (AHEI-2010). Outcomes included coronary heart disease (CHD), hypertension (HT), asthma, cancer (excluding skin cancer), diabetes mellitus (DM), depression and/or anxiety, multimorbidity, and all-cause mortality. Repeated cross-sectional multivariate logistic regressions were performed to investigate the association between baseline DQ and NCDs over 15 years. The AHEI-2010 mean (±sd) for participants (n = 8017) was 51.6 ± 11.0 (range: 19–91). There was an inverse association between AHEI-2010 and incident asthma at survey 4 (OR_Q5–Q1_: 0.75, 95% CI: 0.57, 0.99). Baseline DQ did not predict the occurrence of any NCDs or multimorbidity between the ages of 25–45 years. Further well-planned, large prospective studies conducted in young women are needed to explore dietary risk factors before the establishment of NCDs.

## 1. Introduction

Globally non-communicable diseases (NCDs) were the leading cause of mortality in women in 2019 [1,2]. Coupled with epidemiological transition, changing disease patterns from communicable diseases to NCDs, and demographic transition, resulting in population ageing and growth, NCD occurrence is expected to increase [3]. Furthermore, though NCDs are not restricted to a particular age group or sex, women are more likely to experience their socio-economic impacts compared with men [4]. Of concern, women of childbearing age are susceptible to NCDs, and the most common diseases are cardiovascular disease (CVD), hypertension (HT), cancer and diabetes mellitus (DM) [5]. Some health disorders during pregnancy are associated with NCDs in later life. For instance, hypertension during pregnancy (HTpreg) can affect the vascular health of the mother, thereby leading to the occurrence of CVD and stroke in later life [6,7]. In addition, gestational diabetes mellitus (GDM), one of the hyperglycemic disorders in pregnancy, increases the risk of developing DM in later life [8].

NCD and multimorbidity (the concurrence of two or more chronic diseases in the same person) [9,10] have become public health challenges. Australia is experiencing a catastrophic level of NCDs [11], and estimates from the 2017-18 National Health Survey indicated that almost half of Australian females were experiencing one or more NCDs, including asthma, cancer, DM, and mental and behavioural problems [12]. Notably, 43% of Australian women aged 45 years and younger reported as having at least one NCD [12].

With regard to disease aetiology, a differential distribution of NCD risk factors among men and women was observed in a European study [13]. It was accepted that a set of risk factors such as lifestyle or behavioural factors, environmental factors, underlying metabolic/physiological factors and infections are multifactorial causes that are related to NCD mortality and morbidity [14]. Of these, diet has been identified as one lifestyle or bahavioural risk for CVD, some cancers and DM [15]. Moreover, dietary risk factors are globally ranked second (to high blood pressure) for attributable deaths for women [16]. Diet, the combination of foods and nutrients, can be condensed into a simple measure or summative score [17,18,19] by constructing diet quality indices (DQIs) based on dietary guidelines or specific dietary pattern recommendations [20,21,22]. While this method also has limitations and measurement errors [23], it goes further than the single nutrient approach in recognising the synergistic nature of micro- and macronutrients [24].

A higher score on a DQI generally reflects healthier or optimal diet quality (DQ) or closer compliance to dietary recommendations [25]. The original DQIs such as the Healthy Eating Index (HEI), Diet Quality Index, and Diet Quality Index-Revised were developed in alignment with the Dietary Guidelines for Americans [26,27,28], whereas the Mediterranean Diet Score was based on the Mediterranean dietary pattern [29]. More recently, DQIs have been developed in alignment with country-specific dietary guidelines (e.g., the Dietary Guideline Index [30], China Dietary Guideline Index [31], and Healthy Dietary Habits Index [32]), or modified from earlier indices (e.g., Recommended Food Score [33], Australian Recommended Food Score [34], and alternative Mediterranean Diet [35]). Additionally, the Alternative Healthy Eating Index (AHEI), modelled on the original HEI, was constructed according to extensive epidemiological evidence for NCD prevention [33]. Although both HEI and AHEI are measurements of DQ, there are differences in scoring (e.g., the original HEI included total fat and protein (meat and beans) for fat and protein components [26], whereas the AHEI included trans-fat and the ratio of polyunsaturated fatty acid (PUFA) to saturated fatty acid (SFA) in the fat component; and the ratio of white to red meat, nuts and soy protein in the protein component [33]. Both the HEI and AHEI have been regularly updated (HEI-2005 [36], HEI-2010 [37], HEI-2015 [38], and HEI-2020 [39] for HEI; AHEI-2010 [40] for AHEI). 

DQIs have been extensively applied in diet-health outcome relationship studies, including those considering NCDs [17,19,20,41,42,43,44,45,46,47]. The most commonly assessed health outcomes have been CVD or coronary heart disease (CHD) [41,44], HT [45], cancer [41,42], DM [41,43], depression [46,48], and all-cause mortality [41,47]. Previous evidence has shown that a high level of DQ assessed by the AHEI and/or AHEI-2010 is associated with a reduced risk of CVD, cancer, DM, neurodegenerative disease, and all-cause mortality [41]. However, in term of the clustering of NCDs, evidence on the relationship between overall diet and NCD multimorbidity is very limited [49]. Previously, we examined the relationship between DQ (measured as Healthy Eating Index for Australian Adults-2013 (HEIFA-2013), Mediterranean Diet Score (MDS), and Alternative Healthy Eating Index-2010 (AHEI-2010)) and the incidence of NCDs, including multimorbidity among a cohort of women born between 1946 and 1951 that was drawn from the Australian Longitudinal Study on Women’s Health (ALSWH) [50]. Data showed that mid-aged women with the highest DQI quintiles had reduced odds of NCDs in later surveys (9 to 15 years later), and AHEI-2010 was the most sensitive DQI for prediction of NCDs [50].

Of note, taking a life course approach in NCD prevention and control has been introduced in recent years [51]. Despite slow progression and having a long latency in the development of NCDs [14], it is important to tackle risk factors in earlier life stages, transitional stages and critical life stages [52]. Partnering and parenting are major transitional stages and turning points for young adults [53], as they are associated with changes in dietary habits and physical activity [54]. In a 6-year follow-up of a nationally representative sample, young Australian women who were living with a partner at baseline survey and those who become partnered during follow-up or remained partnered consumed a relatively healthier diet compared to singles [55]. Parenting or living with children has been reported as having favourable and unfavourable effects on dietary intakes or quality [53,55,56,57,58].

Some risk factors for NCDs are prevalent in young women of childbearing age, for example, overweight or obesity [59]. Overweight, obesity, or weight gain in childbearing women is partly associated with marriage, pregnancy and motherhood [59], which could be related to changes in dietary behaviours and/or physical activity during such a transition period [59]. Overweight or obesity is linked to the occurrence of CVD [60], breast cancer and endometrial cancer [61], and DM [62].

The tracking of lifestyle or behavioural factors, especially unhealthy diet, in young women could be a useful strategy for NCD prevention. Given that the prevalence of NCDs is increasing with age, with women disproportionately affected [11,63], and almost half of Australian women who reported NCDs being of childbearing age [11,12], an investigation of DQ as a key modifiable predictor of NCDs among younger women is warranted.

It would be beneficial to make comparisons across generations in investigating the relationship between DQ and NCDs. A longitudinal analysis investigating the temporal associations between DQ and NCDs is needed. However, prior to the initiation of longitudinal analysis, initial cross-sectional analyses using baseline variables would provide context. This study was aimed to determine if DQ, assessed by AHEI-2010, is predictive of NCD outcomes during a 15-year follow-up of women born 1973–78 from the age of 25–30 years. The NCD outcome of interests included CHD, HT, asthma, cancer (except skin cancer), DM, depression and/or anxiety, and multimorbidity. We hypothesized that women with a high DQ assessed by AHEI-2010 would have reduced odds of NCD outcomes and multimorbidity during the 15-year follow-up compared to those with a low DQ.

## 2. Materials and Methods

### 2.1. Study Population

Data for this study were drawn from the Australian Longitudinal Study on Women’s Health (ALSWH), a national population-based study funded by the Commonwealth Department of Health and Ageing. The ALSWH study commenced in 1996 and approximately 45,000 women born between 1973 and 1978, 1946 and 1951, and 1921 and1926 were selected from the Medicare health insurance database, which included all Australian citizens and permanent residents [64]. A new cohort of women born between 1989 and 1995 were enrolled by means of in-person, internet and social media contact methods [65]. Ethical clearance for all participants was granted from the Human Ethics committees of the University of Newcastle and the University of Queensland. Further details about the ALSWH can be found elsewhere [66].

The present study included data from a cohort of women born between 1973 and 1978. A total of 14,247 women aged 18–23 years responded to survey 1 (S1, 1996), then followed up every 3 years (apart from the 4 years between S1, 1996 and S2, 2000) until 2018. Self-administered questionnaires were sent to collect information on women’s physical and mental health, health service use, and socio-demographic and behavioural characteristics. At S3 (aged 25–30 years in 2003), the dietary intakes of women were assessed by the Dietary Questionnaire for Epidemiological Studies version 2 (DQES-v2) [67]. The DQES-v2 has been validated among young Australian women and demonstrated its ability to assess habitual intake (energy-adjusted correlation coefficients: 0.28 to 0.70) [68].

This analysis comprised women at S3 (2003, aged 25–30 years), when dietary data were first assessed. Women were included if they responded to S3 with complete FFQ data and responded at least once between S4 (2006, aged 28–33 years) and S8 (2018, aged 40–45 years). To ensure that the dietary predictor was measured before the onset of the NCD (i.e., incident cases), women were excluded if they reported pre-existing NCDs prior to S3. An exception was made for asthma and depression and/or anxiety, where histories of these conditions were adjusted for and a recent episode was deemed an incident occurrence. Exclusions comprised the self-reported diagnosis of: CHD (heart disease) at S1, S2 and S3; HT (HT (high blood pressure) at S1, HT (high blood pressure) other than during pregnancy at S2 and S3); cancer at S1, S2 and S3; and DM (diabetes (high blood sugar) at S1, non-insulin dependent (type II) diabetes at S2 and S3). Women who had no or missing S3-FFQ data were excluded. Figure 1 shows a simple illustration of the ALSWH 1973–78 data for analysis. In the present analysis, 8017 women from S3 (2003) and afterwards were included (Figure 2). The number of respondents in five consecutive surveys (from S4, 2006 to S8, 2018) can be seen in Appendix A.

### 2.2. Dietary Intake Assessment

The diet of the ALSWH 1973–78 cohort was assessed by using the DQES-v2 [67] at S3 (in 2003). The FFQ included questions for reporting the usual frequency of 74 food items and six alcoholic beverages in the last 12 months. The responses ranged from 1 to 10 points; “Never” to “≥ 3 times per day” for food items and “Never” to “Every day” for beverages. Portion-size photographs were used to adjust respective servings. Further questions assessing the daily servings of fruit, vegetable, bread, dairy products, egg, fat spread, and sugar were also asked. The partaking of food items (in grams/day) and nutrients was calculated using the national food composition database of Australian foods, the NUTTAB95 [69].

### 2.3. Exposure Variable

The exposure variable of the ALSWH 1973–78 cohort was DQ assessed by the Alternative Healthy Eating Index-2010 (AHEI-2010) calculated from the DQES-v2 at S3. Dietary data at S3, assuming constancy throughout the 15-year period, were used in this cross-sectional study to investigate the association between the baseline DQ of the ALSWH 1973–78 cohort and NCDs during follow-ups (from S4 in 2006 to S8 in 2018). Previously, it was shown that the DQ of women in the ALSWH 1946–51 cohort was relatively stable over a 12-year period (from S3, 2001 to S7, 2013) [70]. Therefore, in our ALSWH 1973–78 cohort, DQ at S3 was used to examine the association between DQ and incident NCDs during a 15-year follow up (from S4 to S8). The AHEI-2010 was selected based on its relevance to current dietary recommendations [71], having performed well when critically appraised compared with other DQIs based on a specific dietary pattern [72], and because it allowed us to make comparisons across generations based on our previous ALSWH 1946–51 cohort analyses [50]. The detailed scoring of AHEI-2010 is provided in Appendix A.

The AHEI-2010 was constructed using foods and nutrients for NCD prevention including CVD, some cancers, and DM based on clinical and epidemiological investigations [33,40]. The components of this index were modelled on the original HEI [26] and the original AHEI [33]. Each component ranged from 0 (suboptimal diet) to 10 (optimal diet), and intermediate values were correspondingly assigned. The total score was set as the sum of 11 components so that the AHEI score had a possible range from 0 to 110. The computations of scoring for 11 components have been previously reported in detail [40]. The positive components were vegetables, fruit, whole grains, nuts and legumes, long chain omega-3 fats, and PUFAs. Intakes of sugar sweetened beverages (SSBs) and fruit juice, red and processed meat, trans-fat, and sodium were coded as negative components. Moderate alcohol consumption contributed to a higher score (i.e., higher diet quality).

### 2.4. Outcome Variables

The outcomes of the ALSWH 1973–78 cohort at five surveys (from S4, 2006 to S8, 2018) included the cumulative incidence of NCDs [CHD, HT, asthma, cancer (except skin cancer), DM, and depression and/or anxiety], multimorbidity (the concurrence of two or more of NCDs of interest), and all-cause mortality.

The incidence of each disease was based on the ALSHW survey when it was first reported. The occurrence of common NCDs in the ALSWH 1973–78 cohort was self-reported. At S1, participants were asked if they had a known diagnosis of heart disease, HT (high blood pressure), asthma, cancer, and/or diabetes (high blood sugar) by using the question “Have you ever been told by a doctor that you have…? (circle one number on each line)?” At S2, their known status of heart disease, HT (high blood pressure) other than during pregnancy, asthma, cancer (specify type), non-insulin dependent (type 2) diabetes, and depression (not postnatal) and anxiety disorder was assessed using the question “Have you ever been told by a doctor that you have…? (Mark all that apply).” The responses were “yes, in the last four years” and “yes, more than four years ago”. At S3 and S4, the self-reported diagnosis of heart disease, HT other than during pregnancy, asthma, cancer (specify type), non-insulin dependent (type 2) diabetes, and depression (not postnatal) and anxiety disorder was assessed using the question “In the last three years, have you been diagnosed or treated for…? (Mark all that apply)”. For S5 to S8, the self-reported status of HT, cancer and depression was assessed in terms of HT, cancer (skin cancer and other cancer) and depression by using the same format of S3 and S4. Skin cancer could not be excluded at S4, so cancer (except skin cancer) was used for S5 to S8 only.

The self-reported status of NCDs was considered in terms of enduring conditions, meaning participants who had reported an NCD (except for asthma and depression and/or anxiety) in any survey were counted as having that NCD throughout subsequent surveys (Appendix A). The self-reported diagnosis of NCDs among the ALSWH study participants was considered reliable and valid based on previous evaluations against administrative data [73,74]. The incidence of multimorbidity at five subsequent surveys (from S4, 2006 to S8, 2018) was calculated as the presence of two or more NCDs of interest in any combination. The deaths of ALSWH participants were reported through data linkage with the Australian National Death Index [75]. The Cause of Death file, containing the primary or underlying cause and other additional causes, is available for assessing mortality [75]. For the all-cause mortality of our sample, any death was counted and reported as descriptive data.

### 2.5. Covariates

To obtain unbiased estimates of the association between exposure and outcome, potential confounding variables are adjusted for in statistical analyses [76]. Generally, covariates are selected from possible candidates using statistical prerequisites, criteria for selecting variables (the Directed Acyclic Graph (DAG) based on background knowledge is one approach), and variable selection algorithms [77]. In this study, theoretical models in the form of DAGs [78,79,80] were developed based on background knowledge and literature to identify the relationship between exposure, outcome, and covariates across five surveys (from S4, 2006 to S8, 2018) (Appendix A). Covariates at S3 that were associated with both the exposure and outcomes were considered as confounders and adjusted for in this study. They were variables measuring socioeconomic status (residence status, marital status, education, occupation, and income stress), behaviour (physical activity) and childbearing (history of breastfeeding, history of GDM for DM, and history of HTpreg for HT). The role of prescribed medication in the diet-NCDs pathway differs by condition. For instance, taking cholesterol-lowering agents may prevent CHD, but for DM, participants were more likely to take prescribed medications for diabetes following diagnosis. Therefore, prescribed medicine was not included in DAGs but was included in the sensitivity analysis, testing the effect estimates with and without this variable. Being pregnant at the time of the survey was treated as an indicator variable in the regression models. The body mass index, which can be affected by DQ [81] and can influence NCD outcomes [82,83,84,85,86,87,88], was considered a mediator and not adjusted for in the models.

In this study, participants’ responses for residence status were classified as “major cities, inner regional, and outer regional/rural”. For marital status, responses were classified as “never married, married/de facto, and separated/divorced/widowed”. Education status was classified as “no formal education, high school level, diploma, and university degree”. Occupation was coded as “no paid employment and paid employment”. Participants’ income stress was assessed via questions asking how they were able to manage on available income and classified as “easy and difficult”. Physical activity was measured using the Active Australia Survey items [89] incorporated into the ALSWH surveys by asking two questions on frequency and duration of walking (recreation and transport) and moderate- and vigorous-intensity activity over the previous 7 days. By using responses from these two questions, the metabolic equivalent per minute (MET.mins) was calculated as (3*minutes walking) + (4*minutes moderate activity) + (7.5* minutes vigorous activity). Then, 0–39 MET min/week was classified as “none/sedentary”, 40–599 MET min/week as “low”, 600–1199 MET min/week as “moderate”, and ≥1200 MET min/week as “high” [90]. Childbearing variables were current pregnancy status (no or yes), parity (none or one and above), history of breastfeeding (no or yes), history of GDM (no or yes), and history of HTpreg (no or yes).

The participants’ status of taking prescribed medicine and over-the-counter (OTC) medicine over the previous 4 weeks was reported as the quantity of various medications (S1 and S2) and a binary response (S3 and S4). From S5 onwards, they were asked to record the names of any medication taken over the previous 4 weeks without specifying as prescribed or OTC medicines [91]. We accessed the original medication data for S5 to S8 and created three new variables: taking both prescribed and OTC, prescribed medicine only, and OTC medicine only. Taking prescribed medicine, coded as a binary variable (no or yes), was used in this analysis.

At each ALSWH survey, women were asked “Are you currently pregnant?”. The responses were “yes, no, or don’t know” for S1 to S3 and “no, less than 3 months, 3 to 6 months, more than 6 months, or don’t know” for the remaining surveys. We generated a binary variable for pregnancy status. Moreover, the women who had a child’s date of birth recorded in the child dataset were included as pregnant if they returned a survey 0–9 months before a child’s date of birth.

In the ALSWH child dataset, women who ever had a child were identified based on the date of birth and the breastfeeding status of children. Parity was categorised as “no” and “one and above”. Women who ever had breastfed were recorded in the child dataset. Histories of breastfeeding across the surveys (from S2 to S8) were generated and assigned as a binary variable (no or yes).

In ALSWH main surveys, at S2, women were asked whether they had ever been told that they had GDM and/or HTpreg in the last four years or more than four years ago. At S3 and S4, women were asked to report their diagnosis or treatment status of GDM and HTpreg. From S5 to S7, they were asked to recall the diagnosis or treatment for GDM and HTpreg for each child during their pregnancy. Moreover, in the child dataset, the status of women’s GDM and HTpreg was also recorded. From these two data, binary variables measuring the history of GDM and HTpreg were generated from S2 to S8.

There were missing values in covariates such as the area of residence, marital status, education, occupation, income stress, physical activity, and taking prescribed medicine. The carry-forward approach [92] from the subsequent survey was used from S4 to S7. For example, if the value at S4 was missing, values at adjacent surveys (S3 and S5) were checked and replaced the corresponding value labels. For a missing item at S8, the value at S7 was checked and replaced. After filling, no variable had missing values more than 5% of the total data. However, variables related to childbearing such as current pregnancy status, parity, history of breastfeeding, history of GDM, and history of HTpreg had missing values less than 5%, and the carry-forward approach for missing values was not deemed necessary.

### 2.6. Statistical Analysis

The statistical software package Stata version 15 was used in all analyses. A descriptive summary of AHEI-2010 was reported as a continuous measure (mean ± sd) and categorical measure (quintiles). The baseline characteristics of women (at S3) with respect to the AHEI-2010 quintiles were described as mean± standard deviation (sd) or n (%), and they were compared by using an analysis of variance (ANOVA) or chi-squared test. The occurrences of all-cause mortality and NCDs (each NCD and multimorbidity) were reported. The baseline characteristics of women who had been excluded and included in the current study were compared (Appendix A).

The univariate analyses of AHEI-2010 at S3 and NCD outcomes (each NCD and multimorbidity) over 15 years (from S4 to S8) were performed. The number of CHD events was low compared with other NCDs, so multivariate models were not fitted for the association between AHEI-2010 and CHD. For others, multivariate models (adjusted for covariates at S3 including age, residence, marital status, education, occupation, income stress, physical activity, current pregnancy status, parity, history of breastfeeding, history of GDM for DM, and history of HTpreg for HT) were fitted to investigate the short-term and long-term effects of AHEI-2010 on NCD outcomes. Multivariate logistic regression models [93] were used to investigate the effect of AHEI-2010 on the NCD outcomes. The DQI data from S3, 2003 were used for the prediction of NCD outcomes at five consecutive surveys (from S4, 2006 to S8, 2018). The odds ratios (ORs) and 95% confidence intervals (95% CIs) for NCD outcomes relative to AHEI-2010 were computed using the quintile 1 as the reference category, and they are presented as the main results.

To test the robustness of our results, sensitivity analyses were performed. We performed multiple tests, applying the Bonferroni correction in the logistic regression models to account for the covariates whose 95% CIs were near 1. Analyses that used taking prescribed medicine as a covariate (Appendix A) and changes of childbearing variables (Appendix A) were also performed.

## 3. Results

A total of 8017 women were included at baseline (S3, 2003), with the mean AHEI-2010 score at S3 being 51.6 (sd 11.0; range: 19–91). Women with the AHEI-2010 quintile 5 (Q5 AHEI-2010) were reported as married/de facto, living in major cities, having a university degree, having paid employment, easily managing income, and being more physically active than those in quintile 1 (Q1 AHEI-2010) (Table 1).

The relationship between AHEI-2010 and the risk of common NCDs (each NCD and multimorbidity) are presented in Table 2. Multivariate logistic regression models with and without a history of asthma were fitted. In the models without a history of asthma, women in the Q5 AHEI-2010 had lower odds of asthma in S4 (2006) when compared to those in the Q1 AHEI-2010 (OR: 0.75, 95% CI: 0.57, 0.99). After the Bonferroni correction was applied to the univariate model (asthma at S4) and multivariate model (asthma at S4 without a history of asthma variable) (Table 2), the odds of asthma in S4 (2006) when compared with the highest vs. lowest quintile of AHEI-2010 resulted in a non-significant association (Table 2, footnote: *p*-value = 0.41 and 0.42, respectively). In the model with a history of asthma, the association was attenuated and non-significant. Univariate inverse associations between AHEI-2010 and HT at S7 and S8 were found, though these associations became insignificant in the multivariate models. DQ did not predict the occurrence of DM, cancer, depression and/or anxiety, and multimorbidity (Table 2).

When performing the sensitivity analyses using prescribed medicine as a covariate and changes in childbearing variables, the odds of NCDs and multimorbidity remained consistent (Appendix A).

## 4. Discussion

In this large population-based study of Australian women aged 25–30 years at baseline, greater adherence to the AHEI-2010 was only associated with the occurrence of asthma at 3 years after the measure of diet. Overall, there was no association between the AHEI-2010 and NCDs and multimorbidity. This suggests that a specific dietary pattern, based on clinical and epidemiological evidence and representing dietary components that have been associated with a lower risk of chronic diseases, is not immediately and obviously associated with NCDs in this group of women.

In the present study, we found no association between AHEI-2010 and CHD in univariate analysis. This finding is inconsistent with the previous studies. In an analysis conducted among Caucasian nurses aged 38–63 years at baseline, those in the Q5 AHEI-2010 had a reduced risk of CHD compared to those in the Q1 AHEI-2010 (HR: 0.66, 95% CI: 0.58, 0.78) over 24 years [40]. The inverse association between AHEI-2010 and incident CVD over 22–26 years, including CHD has been supported in US community-based Atherosclerosis Risk in Communities (ARIC) study participants (aged 45–64 years, 56% women at baseline) [94,95]. Furthermore, in a 10-year follow-up of the NutriNet-Sante (NNS) study cohort (mean age = 41 years, 79% women), participants in quartile 4 of AHEI-2010 had a 22% reduced risk of CVD compared to those in quartile 1 [96]. One reason the results may be inconsistent is that the samples in previous research have generally been older than the sample in the current study. DQ quintiles and its distributions reported in previous studies were higher than those of [95] or similar to our study [40,94]. Many components included in AHEI-2010 are foods and nutrients that have been shown to prevent CVD, including CHD [40]; however, we could not investigate the relationship between AHEI-2010 and CHD because of the very low incidence in this cohort. The self-reported prevalences of CHD in Australian women at various age groups were 0.3% at 18–44 years, 0.7% at 45–54 years, 3.6% at 55–64 years, 6.0% at 65–74 years, and 8.1% at 75 years and older [97].

There was no association between AHEI-2010 and HT in multivariate models. HT, as a risk factor of CVD and investigated as raised systolic blood pressure (SBP) or diastolic blood pressure (DBP), has been investigated in NCD-related research [98,99,100]. A significant reduction in SBP, not DBP, was demonstrated among people in quartile 4 of AHEI-2010 compared to those in quartile 1 of the 2007–2010 National Health and Nutrition Examination Survey (NHANES) (n = 4097, aged ≥ 20 years) [100]. Furthermore, there is now considerable evidence on food and nutrients beneficial for blood pressure (BP) or HT such as fruits [101], vegetables [101], legumes [102], omega-3 fatty acids and PUFAs [103]. Nevertheless, there was no association between AHEI-2010 and HT or BP in a nested case-control study conducted within the Singapore Chinese Health Study (SCHS) (n = 1994, aged 45–75 years, 35% women) [99].

In the present study, women in the Q5 AHEI-2010 had a 25% reduction in the odds of asthma at S4 (2006) compared to those in the Q1 AHEI-2010. However, the association was attenuated and was not found to be statistically significant after applying a Bonferroni correction in the multivariate logistic regression models. Current evidence on DQ and asthma outcome is inconclusive, and studies investigating the role of diet in adults have been limited [104]. The preventive roles of fruits, vegetables, and vitamin E [105,106], as well as fiber [107], and the deleterious roles of red processed meat [108,109] and SSB [110] on asthma have been documented in observational studies. A favourable effect of DQ on asthma was observed in the French prospective Epidemiological study on the Genetics and Environment and Asthma Study (n = 969, mean age = 43 years, 51% women), indicating that a 10-unit increase in AHEI-2010 was shown to improve asthma symptoms, measured as the frequency of respiratory symptoms during the last year [111]. In another French cohort study (n = 26,197 women, aged ≥ 18 years), a 21% reduced odds of asthma symptoms in women and AHEI-2010 was also observed [112]. Opposite findings were observed among 73,228 women from NHS [113] and among 12,687 adult participants (60.2% women) from the Hispanic Community Health Study/Study of Latinos [114]. The conflicting findings between studies could be related to different methodological approaches of measurements for asthma such as asthma symptom score [111,112], self-reported diagnosis [113], current asthma status [114] and for dietary assessments such as FFQ [111,113] or 24-hour dietary recalls [112,114].

With regard to overall cancer and DQ, an inverse association between AHEI-2010 and overall cancer was observed in 71,495 women aged 38–63 years from the NHS over 24 years [40] and 41,543 participants from the NNS study over 9 years (aged ≥40 years, 73.5% women) [115]. In our study, we did not find any association between DQ and cancer (excluding skin cancer). The previous analysis conducted among the ALSWH cohort showed similar results [50]. The discrepancies between studies might be partly explained by shorter follow-up durations since cancer cannot be detected within a far shorter period [116]. Another reason may be the components measured in AHEI-2010 were not cancer-specific but based on dietary factors related to chronic disease risk in general [40]. Given that specific types of fruits and vegetables have an impact on cancer, their effect could not be found when all fruits and vegetables are combined as components in a DQI [117]. Furthermore, the endpoint for total cancer is heterogeneous compared with other NCDs such as CHD, asthma, or DM [40].

Previous studies have demonstrated a significant inverse association between AHEI-2010 and the risk of DM among participants from the NHS over 24 years of follow-up (n = 71,495, aged 38–63 years) [40], among white women from the MEC study (aged 45–74 years, 53% women) [118], among participants from the Women’s Health Initiative Observational Study (aged 50–79 years) [119], among women from the Singapore Chinese Health Study (aged 45–74 years) [120], and among the ALSWH 1946–51 cohort (n = 5350, aged 50–55 years) [50]. However, these findings were not replicated in the present study or in a recent analysis of the ARIC Study over a median follow-up of 22 years (n = 10,808, aged 45–64 years, 56.1% women) [95]. Many components in AHEI-2010 had been constructed based on the optimal dietary factors for the prevention of DM [40]. Notwithstanding AHEI-2010 being the most sensitive DQI for incident DM among the three DQIs tested in the previous ALSWH cohort analysis [50], these results cannot extend to the younger age ALSWH cohort.

Previous evidence showed that healthy dietary patterns characterized by vegetables, fruits and whole grains was inversely associated with depression [121]. The underlying preventive effects of vegetables and fruits for depression may be related to (1) the antioxidant actions of vitamin C, vitamin E, and carotenoid compounds; (2) the balancing of neurotransmitter levels, for instance, by reducing homocysteine concentrations by folates [122]. The anti-inflammatory properties of long-chain omega-3 PUFA also contribute to neurotransmission and are beneficial for depression [123]. The deleterious effects of red/processed meat and SSBs have been shown in previous studies [124,125]. In the scoring of AHEI-2010, higher points are given to high intakes of beneficial components and low intakes of deleterious components. The preventive potential of DQ assessed by AHEI-2010 on depression and/or anxiety was observed among Iranian adults (n = 3363, mean age = 36 years, 58.3% women) [126] and Spanish adults (n = 15,093, aged ≥18 years, 59% women) [127]. However, there was no association between AHEI-2010 and depression and/or anxiety in the present study, nor in a French cohort (n = 26,225, aged 18–86 years, 76% women) [128]. Compared with our results, reported prevalence or incidence of depression were higher in the Iranian study (30% in total participants, 35% in women) [126] but lower in the Spanish (~7%) [127] and French (~8.3%) [128] studies. Given that the incidence of depression and/or anxiety in our participants (~15%) was lower than Iranian women (35%), it might have been underpowered to detect the difference.

Regarding the relationship between dietary factors and NCD multimorbidty, the literature is limited and inconclusive [50,129,130,131,132,133,134,135,136]. Inverse associations between vegetable and fruits consumption and NCD multimorbidity have been documented in cross-sectional [130,131] and longitudinal [132,133,134] studies. The harmful effect of soft drink consumption on multimorbidity was reported in an Australian study (n = 36,663, aged ≥ 16 years, 51.1% women) [129]. In exploring the relationship between overall diet assessed by DQIs and multimorbidity, decreased relative odds were observed among European adults who adhered to the Mediterranean diet (n = 291,778, aged 43–58 years, 64% women vs. n = 1140, aged 18 years, 56% women) [135,136] and Australian women who adhered to the Australian Dietary Guideline and AHEI-2010 (n = 5,350, aged 50–55 years) [50] when compared to those who did not. In contrast to the previous findings [50], there was no association between AHEI-2010 and NCD multimorbidity in this study. A potential explanation for the lack of association could be the differences between the AHEI-2010 scores amongst the ALSWH 1946–51 cohort (mean (±sd): 56.0 ± 10.3, range: 26–93.8) and this study cohort (mean (±sd): 51.6 ± 11.0, range: 19–91).

Descriptive summaries of AHEI-2010 have been reported in previous studies: similar to [40,94,119], higher than [50,95,98,118,127], or smaller than [113] the present study. Comparisons cannot made with some studies since they did not include trans-fat [99,100,111,112,115,120] or alcohol [126] because those data were not available.

The application of the most sensitive DQI, AHEI-2010, in predicting NCD outcomes in the previous study [50] is one of the strengths of this study. AHEI-2010 is the latest version based on the clinical and epidemiological evidence, and it is widely used in diet-health relationship studies [41]. However, most of these studies have been conducted in the United States [41]. A few Australian studies have used AHEI-2010 as a measurement of DQ [137,138,139,140,141], and only two studies were related to NCDs such as urothelial cell carcinoma [141] and ovarian cancer survival [137]. Hence, this study adds to the body of evidence on DQ. The analysis of healthy childbearing women from a nationally representative population [142] is another strength of this study.

Several limitations should be acknowledged. Firstly, in comparison with the originally recruited young women at S1 (n = 14,247), those excluded from the study for the reasons of having NCDs and missing FFQ at S3 (n = 1,064) were more likely to be married, originate from inner regional and outer regional/rural areas, have no formal education, have no paid employment, have difficulty in income management, have lower physical activity, have poor/fair self-rated health, and have taken prescription medications (Appendix A). Women excluded from the present study were distributed in lower DQ quintiles such as quintiles 1 and 2 (data not shown). This affected the examination of DQ differences between the groups, making it difficult to detect small variations that could have been biased towards the null. Secondly, the exposure and variables at S3 were used in logistic regression models. Cross-sectional studies cannot determine whether there is temporal relationship between exposure and outcome of interests. A longitudinal analysis that allows for the adjustment of time-varying covariates should be performed. However, the effects of change in childbearing variables such as current pregnancy status, parity, history of breastfeeding, history of GDM, and history of HTpreg were investigated in sensitivity analysis (Appendix A). Thirdly, the insufficient NCD cases at each survey, particularly CHD and DM, could have introduced high uncertainty in our effect sizes. Fourthly, the number of cancer cases (except skin cancer) could have been affected by the inability to exclude skin cancer at earlier surveys (from S1 to S4). However, when the proportions of cancer cases (not specified) at S1 to S4 were checked amongst skin cancer cases across S5 to S8, these data comprised no more than 5% (data not shown). The selected NCDs are characterised by increasing incidence with age, which makes it difficult to obtain results based on clinical endpoints such as CHD and DM. However, looking at modifiable risk factors early, even prior to disease onset, is important given that Australian reports have shown that 43% of women aged 45 and younger are affected by one or more NCDs [11,12]. Exploring NCD risk factors and considering which NCDs are the most prevalent at different life stages [143,144] are important for developing appropriate primary prevention strategies. In assessing dietary intake, FFQs, such as the DQES-v2, are more useful in providing relative ranking than absolute values of intake [145]. Calculating a DQI from FFQs introduces further uncertainty in the dietary measure and by no means represents a complete picture of diet [146]. Instead, it provides a simple summary that performs well across large-scale national and international studies [23]. Lastly, residual confounding is still possible even though appropriate covariates adjustments were made with multivariate analysis and selected based on DAGs.

## 5. Conclusions

A high DQ measured by AHEI-2010 was only associated with the occurrence of asthma at S4. A cross-sectional analysis examining the association between overall DQ and NCDs could not determine the causal relationship. A longitudinal analysis is therefore necessary to investigate the temporal associations. While most research examining diet-health outcome relationship has suggested the preventive effect of overall diet on NCDs, the present findings did not demonstrate early evidence of this between the ages of 25–45 years. Further well-planned, large prospective studies with sensitive indicators of NCD risk factors and incident NCDs are indispensable.

## Figures and Tables

**Figure 1 nutrients-14-04403-f001:**
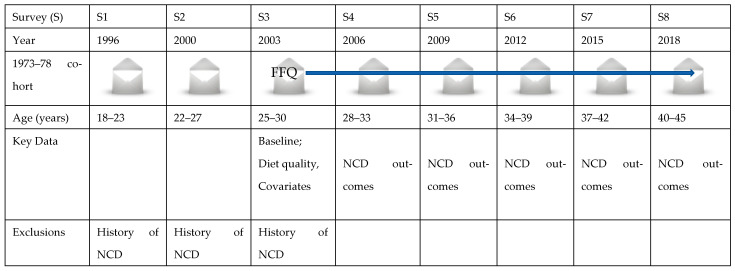
Simplified diagram showing women born between 1973 and 1978 in the Australian Longitudinal Study on Women’s Health (ALSWH).

**Figure 2 nutrients-14-04403-f002:**
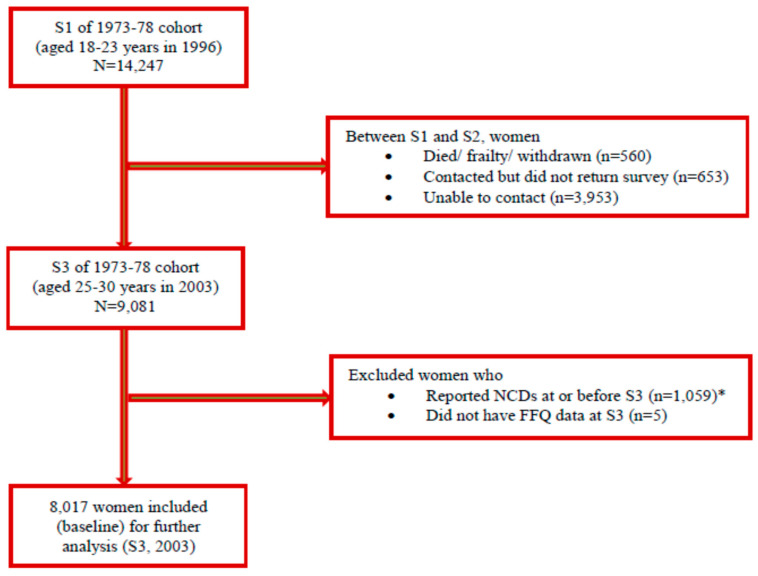
Diagram showing participants’ sampling from the Australian Longitudinal Study on Women’s Health (ALSWH), born in 1973–78. * NCDs at or before S3 were coronary heart disease, hypertension, cancer and diabetes mellitus.

**Table 1 nutrients-14-04403-t001:** Baseline characteristics of the sampled women (n = 8017; survey 3 in 2003) by quintiles of Alternative Healthy Eating Index-2010.

	AHEI-2010 Quintiles					
Characteristics	Q1 (n = 1635)	Q2 (n = 1572)	Q3 (n = 1582)	Q4 (n = 1636)	Q5 (n = 1592)	*p*-Value ^§^
**Age (years) [mean (sd)]**	27.5 (1.5)	27.6 (1.5)	27.5 (1.5)	27.6 (1.5)	27.6 (1.4)	0.02 **
**Marital status [n (%)]**						<0.001 **
Never married	415 (25.5)	487 (31.1)	557 (35.3)	630 (38.6)	734 (46.3)	
Married/de facto	1141 (70.1)	1028 (65.6)	974 (61.7)	939 (57.5)	810 (51.1)	
Separated/divorced/widowed	72 (4.4)	52 (3.3)	47 (3.0)	64 (3.9)	42 (2.6)	
**Area of residence [n (%)]**						<0.001 **
Major cities	810 (49.6)	835 (53.2)	868 (55.0)	992 (60.7)	1002 (63.2)	
Inner regional	499 (30.5)	435 (27.7)	430 (27.2)	388 (23.7)	388 (24.4)	
Outer regional/rural	325 (19.9)	300 (19.1)	281 (17.8)	254 (15.6)	197 (12.4)	
**Education [n (%)]**						<0.001 **
No formal education	21 (1.3)	22 (1.4)	21 (1.3)	11 (0.7)	9 (0.6)	
High school level	582 (36.1)	507 (32.7)	452 (29.0)	375 (23.3)	289 (18.4)	
Diploma	449 (27.9)	414 (26.7)	398 (25.6)	402 (25.0)	340 (21.7)	
University degree	558 (34.7)	607 (39.2)	685 (44.1)	821 (51.0)	929 (59.3)	
**Occupation [n (%)]**						<0.001 **
No paid employment	393 (24.2)	354 (22.6)	269 (17.1)	257 (15.9)	195 (12.3)	
Paid employment	1228 (75.8)	1210 (77.4)	1304 (82.9)	1364 (84.1)	1385 (87.7)	
**Income stress [n (%)]**						<0.001 **
Easy	885 (54.3)	884 (56.4)	904 (57.4)	1031 (63.2)	1036 (65.2)	
Difficult	746 (45.7)	683 (43.6)	670 (42.6)	600 (36.8)	552 (34.8)	
**Physical activity [n (%)]**						<0.001 **
None/sedentary	204 (12.7)	167 (10.8)	129 (8.3)	101 (6.3)	56 (3.6)	
Low	625 (38.9)	578 (37.3)	510 (32.7)	482 (29.8)	375 (23.8)	
Moderate	359 (22.3)	355 (22.9)	380 (24.2)	414 (25.6)	381 (24.2)	
High	420 (26.1)	450 (29.0)	543 (34.8)	618 (38.3)	761 (48.4)	
**Taking prescribed medicine [n (%)]**						0.22
No	1170 (72.7)	1107 (71.5)	1134 (72.7)	1206 (74.6)	1171 (74.5)	
Yes	440 (27.3)	442 (28.5)	425 (27.3)	411 (25.4)	401 (25.5)	

The number of participants in each Alternative Healthy Eating Index-2010 (AHEI-2010) quintile varied because of missing data in the covariates. ** *p*-value < 0.05. ^§^
*p*-values were obtained from analysis of variance (ANOVA) for continuous variables and chi-squared test for categorical variables.

**Table 2 nutrients-14-04403-t002:** Odds of common non-communicable diseases (including multimorbidity) over 15 years of follow-up (from survey 4, S4 to survey 8, S8) based on quintiles of Alternative Healthy Eating Index-2010: 1973–78 Australian Longitudinal Study on Women’s Health cohort.

Survey	S4	S5	S6	S7	S8
**NCD**	**OR (95% CI)**	**OR (95% CI)**	**OR (95% CI)**	**OR (95% CI)**	**OR (95% CI)**
**CHD**	**n = 11**	**n = 17**	**n = 29**	**n = 42**	**n = 69**
	S4 (n = 6871) ^a^	S5 (n = 6127) ^a^	S6 (n = 6017) ^a^	S7 (n = 5452) ^a^	S8 (n = 5394) ^a^
Univariate	1.0 (0.1–7.0)	3.9 (0.4–34.6)	1.2 (0.3–4.3)	2.7 (0.6–13.6)	1.1 (0.5–2.5)
**HT**	**n = 77**	**n = 231**	**n = 346**	**n = 433**	**n = 556**
	S4 (n = 6871) ^a^	S5 (n = 6127) ^a^	S6 (n = 6017) ^a^	S7 (n = 5452) ^a^	S8 (n = 5394) ^a^
Univariate	0.9 (0.4–1.8)	0.7 (0.4–1.1)	0.7 (0.5–1.0)	0.6 (0.4–0.9) *	0.7 (0.5–0.9) *
	S4 (n = 6608) ^b^	S5 (n = 5905) ^b^	S6 (n = 5814) ^b^	S7 (n = 5268) ^b^	S8 (n = 5214) ^b^
Multivariate ^c^	1.0 (0.5–2.3)	0.7 (0.4–1.2)	0.7 (0.5–1.0)	0.7 (0.5–1.1)	0.8 (0.6–1.1)
**Asthma**	**n = 662**	**n = 559**	**n = 558**	**n = 464**	**n = 478**
	S4 (n = 6871) ^a^	S5 (n = 6127) ^a^	S6 (n = 6017) ^a^	S7 (n = 5452) ^a^	S8 (n = 5394) ^a^
Univariate	0.76 (0.59–0.99) * ^§ ¥^	0.8 (0.6–1.0)	0.8 (0.6–1.1)	0.9 (0.6–1.2)	0.9 (0.6–1.1)
	S4 (n = 6621) ^b^	S5 (n = 5914) ^b^	S6 (n = 5824) ^b^	S7 (n = 5279) ^b^	S8 (n = 5226) ^b^
Multivariate ^d^	0.75 (0.57–0.99) * ^§ ¥^	0.8 (0.6–1.0)	0.8 (0.6–1.1)	0.8 (0.6–1.1)	0.8 (0.6–1.1)
Multivariate ^d+^	0.8 (0.6–1.1)	0.8 (0.6–1.1)	0.9 (0.7–1.2)	0.9 (0.6–1.2)	0.8 (0.6–1.2)
**Cancer (excludes skin cancer)**	**n = 68**	**n = 100**	**n = 148**	**n = 200**	**n = 274**
	S4 (n = 6871) ^a^	S5 (n = 6127) ^a^	S6 (n = 6017) ^a^	S7 (n = 5452) ^a^	S8 (n = 5394) ^a^
Univariate	1.3 (0.6–2.8)	1.5 (0.7–2.9)	1.1 (0.6–1.8)	1.0 (0.6–1.6)	0.9 (0.6–1.3)
	S4 (n = 6621) ^b^	S5 (n = 5866) ^b^	S6 (n = 5785) ^b^	S7 (n = 5279) ^b^	S8 (n = 5226) ^b^
Multivariate	1.4 (0.6–3.1)	1.4 (0.7–3.0)	1.1 (0.6–2.0)	0.9 (0.5–1.5)	0.85 (0.6–1.3)
**DM**	**n = 24**	**n = 62**	**n = 110**	**n = 136**	**n = 167**
	S4 (n = 6871) ^a^	S5 (n = 6127) ^a^	S6 (n = 6017) ^a^	S7 (n = 5452) ^a^	S8 (n = 5394) ^a^
Univariate	0.8 (0.2–3.0)	1.4 (0.5–3.6)	0.8 (0.4–1.6)	0.7 (0.4–1.4)	0.6 (0.3–1.1)
	S4 (n = 6560) ^b^	S5 (n = 5905) ^b^	S6 (n = 5814) ^b^	S7 (n = 5268) ^b^	S8 (n = 5214) ^b^
Multivariate ^e^	0.9 (0.2–4.0)	1.5 (0.5–4.5)	0.8 (0.4–1.6)	0.7 (0.3–1.4)	0.6 (0.3–1.3)
**Depression and/or anxiety**	**n = 999**	**n = 1106**	**n = 1199**	**n = 1058**	**n = 1024**
	S4 (n = 6871) ^a^	S5 (n = 6127) ^a^	S6 (n = 6017) ^a^	S7 (n = 5452) ^a^	S8 (n = 5394) ^a^
Univariate	1.0 (0.8–1.2)	1.0 (0.8–1.2)	1.0 (0.8–1.2)	0.9 (0.8–1.2)	0.9 (0.8–1.2)
	S4 (n = 6621) ^b^	S5 (n = 5914) ^b^	S6 (n = 5824) ^b^	S7 (n = 5279) ^b^	S8 (n = 5226) ^b^
Multivariate ^f^	1.0 (0.8–1.3)	1.0 (0.8–1.2)	1.1 (0.9–1.4)	1.0 (0.8–1.3)	1.0 (0.8–1.3)
**Multimorbidity**	**n = 198**	**n = 253**	**n = 360**	**n = 346**	**n = 413**
	S4 (n = 6871) ^a^	S5 (n = 6127) ^a^	S6 (n = 6017) ^a^	S7 (n = 5452) ^a^	S8 (n = 5394) ^a^
Univariate	0.9 (0.6–1.5)	0.9 (0.6–1.4)	0.9 (0.6–1.2)	1.0 (0.7–1.4)	0.9 (0.6–1.2)
	S4 (n = 6621) ^b^	S5 (n = 5914) ^b^	S6 (n = 5824) ^b^	S7 (n = 5279) ^b^	S8 (n = 5226) ^b^
Multivariate ^f^	1.1 (0.7–1.8)	0.9 (0.6–1.4)	0.9 (0.6–1.3)	1.0 (0.7–1.5)	0.8 (0.6–1.2)

CHD: coronary heart disease; CI: confidence interval; DM: diabetes mellitus; HT: hypertension; NCD: non-communicable disease; OR: odds ratio; S: survey. OR (95% CI) expressed in the table is the comparison of odds of having NCDs (each disease and multimorbidity) in the highest quintile to the lowest quintile of AHEI-2010. The main presented results were ORs (95% CI) obtained from respective models. ORs (95% CIs) in respective models were rounded off to 1 decimal place. ^¥^ ORs (95% CIs) were expressed in 2 decimal places when the CI was close to 1 and rounding would have altered the interpretation. The Bonferroni correction was applied to logistic regression models where 95% CIs were near 1. ^§^ Inverse association became non-significant after Bonferroni correction was applied (*p*-value = 0.41 for univariate model and *p*-value = 0.42 for multivariate model without a history of asthma variable). Age, socioeconomic status (marital status, residence, education, occupation, and income stress), and the behavioural variable (physical activity) were adjusted for in multivariate models of all NCD outcomes. ^a^ Number in bracket shows the number of participants in respective surveys for univariate regression; ^b^ Number in bracket shows the number of participants in respective surveys for multivariate regression; ^c^ A history of hypertension during pregnancy at S3 was included as a covariate; ^d^ Model without a history of asthma at S3; ^d+^ Model with a history of asthma at S3; ^e^ A history of gestational diabetes mellitus at S3 was included as a covariate; ^f^ A history of depression and/or anxiety at S3 was included as a covariate. Texts in bold and italic represent accumulative figures of NCD cases (except asthma, depression and/or anxiety) in every survey. * *p*-value < 0.05.

## Data Availability

ALSWH survey data are owned by the Australian Government Department of Health and due to the personal nature of the data collected, release by ALSWH is subject to strict contractual and ethical restrictions. Ethical review of ALSWH is by the Human Research Ethics Committees at The University of Queensland and The University of Newcastle. De-identified data are available to collaborating researchers where a formal request to make use of the material has been approved by the ALSWH Data Access Committee. The committee is receptive of requests for datasets required to replicate results. Information on applying for ALSWH data is available from https://alswh.org.au/for-data-users/applying-for-data/ (accessed on 7 March 2021).

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
