# Peer review of "Alternative Healthy Eating Index-2010 and Incident Non-Communicable Diseases: Findings from a 15-Year Follow Up of Women from the 1973–78 Cohort of the Australian Longitudinal Study on Women’s Health"

_nutrients, 2022, doi:10.3390/nu14204403_

Round 1

Reviewer 1 Report

A small database, quite a lot of information brought in. However, some elements of Prague are very common, handing out financial slogans. If the authors of the pavement look at the table, you will get the analysis yourself. In the opinion of the reviewer, the journal would rewrite the article to make the website more characterful and not commonplace.

Author Response

We have not made any amendments in response to these comments, which do not appear to correlate in any way with our submission. We found these comments confusing, and it raises questions about whether the reviewer is suitably qualified for academic peer review.

Reviewer 2 Report

The study investigates whether DQ measured by AHEI-2020 predicted NCD, including multimorbility, in women from the ALSWH born between 1973-78.

In line 30, the authors say that NDC is the leading cause of mortality in women in 2019. Where? In the world? In Australia?

In line 37, "Task Force on Women..." is quoted. In the bibliography, it should be specified that it is: an article, a web page,  ..

At line 57, the authors state that "a single nutrient approach becomes challenging in examining the relationship between diet...". They then say that "diet can be condensed into a measure..." These two sentences contradict each other.

On line 158, the authors indicate that diet data were obtained in S3(2003) and S5(2009). And on line 169 that was with S3. What is correct?

On line 192, the authors indicate that alcohol consumption was considered a healthy component if consumed in moderation. What criteria have you used to establish what moderation is?

In section 2.5 the calculation of "confounders" is discussed, and the corresponding graphs are presented. But in the results, it is not clear how these "confounding" variables have been eliminated from each of the analyses.

In the statistical analysis section, it is mentioned that continuous and categorical measures of AHEI-2010 have been used. In which analysis have continuous measures of AHEI-2010 been used? And in the analysis where it has been used, it must be indicated.

Line 319 indicates that the women with the highest AHEI-2010 score, which I assume will be Q5, are characterized by the fact that they have never been married. If you refer to Q5, the truth is that the highest percentage occurs in "married/de facto" (51.1%).

Why has the Bonferroni correction not been applied? Since the OR values are so close to 1, it is possible that if it had been applied there would be no significant difference.

Regarding the non-coincidence with other studies in relation to CHD, the authors do not state that all the references they present are studies of women older than those in their sample.

Regarding the limitations, the authors indicate that a limitation is that the same participants have indicated their diseases. But then they say that the administration data validates them. Then it is no longer a limitation and should not be indicated in this section.

One limitation is that it is a “cross-sectional study” but in line 172 they indicate that they have the data to carry out a longitudinal study. Given that the results obtained contradict what the bibliography says, shouldn't they have presented the longitudinal study?

The authors do not assess the possibility that AHEI-2010 is not the appropriate tool to establish the association between NCDs and DQ. Nor do they value that the age group used is appropriate. The authors indicate the rates of the diseases evaluated in Australian women and they are low. In addition, the chosen diseases are characterized by increasing their incidence with age, which makes it much more difficult to obtain results.

Round 2

Reviewer 2 Report

The authors have accepted all proposed changes. And you have justified the doubts that have been presented to them.

Author Response

Thank you very much for your review.
